# Iron Deficiency as a Therapeutic Target in Cardiovascular Disease

**DOI:** 10.3390/ph12030125

**Published:** 2019-08-28

**Authors:** Samira Lakhal-Littleton

**Affiliations:** Department of Physiology, Anatomy & Genetics, University of Oxford, Parks Road, Oxford OX1 3PT, UK; samira.lakhal-littleton@dpag.ox.ac.uk

**Keywords:** iron, hepcidin, iron regulatory proteins, cardiomyocyte, chronic heart failure, pulmonary arterial smooth muscle cells, pulmonary arterial hypertension

## Abstract

Iron deficiency is the most common nutritional disorder in the world. It is prevalent amongst patients with cardiovascular disease, in whom it is associated with worse clinical outcomes. The benefits of iron supplementation have been established in chronic heart failure, but data on their effectiveness in other cardiovascular diseases are lacking or conflicting. Realising the potential of iron therapies in cardiovascular disease requires understanding of the mechanisms through which iron deficiency affects cardiovascular function, and the cell types in which such mechanisms operate. That understanding has been enhanced by recent insights into the roles of hepcidin and iron regulatory proteins (IRPs) in cellular iron homeostasis within cardiovascular cells. These studies identify intracellular iron deficiency within the cardiovascular tissue as an important contributor to the disease process, and present novel therapeutic strategies based on targeting the machinery of cellular iron homeostasis rather than direct iron supplementation. This review discusses these new insights and their wider implications for the treatment of cardiovascular diseases, focusing on two disease conditions: chronic heart failure and pulmonary arterial hypertension.

## 1. Iron and Heart Function

Iron deficiency is defined as reduced iron availability due to depleted iron stores (e.g., pregnancy), inadequate dietary iron intake (e.g., malnutrition), repressed iron absorption (e.g., inflammatory setting), or iron loss (e.g., menstrual blood loss in women of reproductive age). It has a significant negative impact on the quality of life. It affects intrauterine development, reduces physical and mental exercise capacity, and increases morbidity when concurrent with cardiovascular diseases, such as chronic heart failure (CHF), acute heart failure (AHF), pulmonary arterial hypertension (PAH), chronic obstructive pulmonary disease (COPD), and abdominal aortic aneurysm (AAA) [1,2,3,4,5,6,7,8]. The aetiology of iron deficiency in cardiovascular diseases is not entirely clear, but evidence supports the role of inflammation, and particularly of the inflammatory cytokine interleukin-6 in blocking iron absorption in the gut [9,10].

### 1.1. Effects of Iron Deficiency on Heart Function

In the heart, iron is present predominantly in ferritin and within the mitochondrial compartment [11,12]. The latter is in the form of iron–sulphur clusters and heme functional groups [13]. Iron in iron–sulphur and heme groups is required for electron transfer and oxygen activation in oxidative phosphorylation [14,15,16]. Additionally, labile-free iron is required for oxygen activation by dioxygenases [17,18], and as a catalyst for the production of reactive oxygen species (ROS) which are essential for redox signalling [19,20].

Iron deficiency is the most common cause of anaemia [21]. Anaemia indirectly affects cardiac function by reducing muscle exercise capacity, and directly by limiting oxygen availability for use in oxidative phosphorylation within cardiomyocytes [22,23,24]. As well as limiting iron availability to the bone marrow, iron deficiency also limits the iron that is available for uptake by non-erythroid tissues. Recently, the direct effects of limited iron availability in cardiomyocytes were demonstrated in a series of experimental models. These models were designed to generate iron deficiency in the heart while maintaining intact systemic iron homeostasis. In a mouse model lacking cardiomyocyte transferrin receptor 1 (TfR1), iron levels in the heart were severely reduced, resulting in fatal heart failure in the second week of age, in part, due to failure of mitochondrial respiration [25]. Another mouse model of cardiac iron deficiency, achieved through increased iron export in cardiomyocytes, also resulted in heart failure, likely explained by a decrease in the activity of the electron transport chain [26]. Studies of dietary iron restriction in animal models have also shown reduced cardiac contractility due to impaired calcium handling in the iron-deficient hearts [27]. The effects of iron deficiency on calcium handling recapitulated the changes induced by hypoxia in cultured cardiomyocytes, supporting the notion that they are driven by reduced oxygen delivery in anaemic animals [27]. However, the direct effects of iron deficiency on hypoxic signalling in the heart could also be important in this context. Indeed, oxygen sensing by hypoxia-inducible factor (HIF) prolyl hydroxylase (PHD) is modified by intracellular iron availability, due to the requirement of PHDs for iron as a co-factor [28]. In summary, iron deficiency affects heart function through anaemia-dependent and anaemia-independent mechanisms. These mechanisms involve changes in oxidative phosphorylation, calcium handling, and oxygen sensing.

### 1.2. Treating Iron Deficiency in Heart Failure—Current Approaches

Iron deficiency commonly co-exists with cardiovascular disease [4,5,6,7,8,29,30]. It is an established co-morbidity in both chronic and acute heart failure [4,5]. Some studies have reported iron deficiency in up to 50% of patients with heart failure [29,30]. Regardless of the presence or absence of anaemia, iron deficiency is associated with higher mortality, adverse cardiovascular events, and reduced quality of life in these patients [4,5,30]. In recent years, a number of trials have explored whether iron supplementation could be beneficial in patients with chronic heart failure. Oral iron supplements have had limited success, and this has been attributed to suboptimal absorption due to inflammation. However, intravenous iron preparations have shown clear benefits. For instance, ferric carboxymaltose and iron sucrose have both been shown to improve many outcomes in patients with chronic heart failure, such as mortality, self-reported patient global assessment, exercise capacity, hospitalization due to cardiovascular events, and mortality [31,32,33]. Consequently, intravenous iron replacement in chronic heart failure is now recommended by the European Society of Cardiology [34]. The benefits of iron supplementation in acute heart failure are less clear. One study reported that iron supplementation within four days of myocardial infarction (MI) improved infarct healing, and left ventricular(LV) remodelling at a three-month follow-up [35], while another found that high levels of serum iron correlated with short-term mortality following MI [36]. These conflicting results may simply reflect distinct and possibly opposing effects of iron availability on the extent of ischemia reperfusion injury and on subsequent cardiac repair.

### 1.3. Treating Iron Deficiency in Heart Failure—New Approaches

The beneficial effects of intravenous iron supplementation in patients with chronic heart failure likely depend on a combination of mechanisms, including an improvement in exercise capacity and oxygen delivery to tissues. Animal studies have also shown that such interventions can directly correct cardiac iron deficiency [26]. Myocardial iron content has been shown to be reduced in patients with heart failure, where it is shown to be associated with reduced activity of the citric acid cycle enzymes aconitase and citrate synthase [37]. Thus, correction of myocardial iron deficiency may well constitute an additional mechanism through which iron interventions improve outcomes in CHF. However, these iron preparations have been developed to correct anaemia. New approaches need to be considered to directly target myocardial iron deficiency, using compounds that can safely and efficiently replenish cardiomyocyte iron levels. In this context, avoidance of iron toxicity in the heart should be an important consideration. Indeed, the heart’s function is particularly sensitive to excess iron accumulation, as demonstrated by the iron overload cardiomyopathy that occurs in patients with iron overload conditions such as hemochromatosis and thalassemia [38,39]. In line with this, a mouse model harbouring a cardiomyocyte-specific deletion of the iron export protein ferroportin (FPN) was also found to develop fatal cardiomyopathy as a consequence of cardiomyocyte iron-overload [40].

An alternative approach to supplementing iron to the heart is to target the molecular machinery of cellular iron homeostasis in cardiomyocytes. Like other cells in the body, cardiomyocytes utilise the iron regulatory proteins (IRPs) to orchestrate iron uptake, utilisation, and storage. Mice with the cardiac-specific deletion of IRPs fail to increase LV systolic function in response to the dobutamine challenge and have worse LV dysfunction and higher mortality following myocardial infarction [41]. At the same time, IRP activity has been shown to be reduced in the failing human heart [41]. Together, these findings support the notion that myocardial iron deficiency may develop as a consequence of reduced IRP activity in the failing heart, and that this, in turn, impairs cardiac reserve and response to injury. Enhancement of the IRP pathway, by increasing the expression or stability of IRPs, may, therefore, present novel therapeutic strategies in heart failure. The advantage of such an approach over systemically administered iron compounds is that it may enable a more measured correction of intracellular iron levels within cardiomyocytes, reducing the risk of iron toxicity.

Recently, another mechanism of cellular iron homeostasis was identified, in which cardiomyocytes utilise hepcidin, also known as human antimicrobial peptide (HAMP), in an autocrine manner to regulate the iron export protein FPN. Despite maintenance of normal systemic iron levels, deletion of hepcidin specifically in cardiomyocytes resulted in fatal LV failure in mice [26]. Cardiomyocyte-specific knock-in of the FPN isoform C326Y, which retains its iron export function but loses its hepcidin binding, also resulted in heart failure of a similar nature and time-course to that seen in animals lacking cardiomyocyte hepcidin [26]. In both mouse models, increased FPN-mediated iron export caused the depletion of iron from cardiomyocytes. Heart failure in cardiac hepcidin-knockouts could be prevented by intravenous iron supplementation. Additionally, animals with hepcidin-deficient hearts developed greater hypertrophic response to sustained dietary iron restriction than their littermate controls, indicating that the cardiac hepcidin/FPN axis functions to protect the heart in the setting of systemic iron deficiency [26]. These studies identify the cardiac hepcidin/FPN axis as a non-redundant component of cellular iron homeostasis in cardiomyocytes, and further show that the control of the intracellular iron pool in these cells is required for normal cardiac function [26]. Studies comparing mice harbouring tissue-specific with those harbouring a ubiquitous disruption of the hepcidin/FPN axis support the notion that intracellular iron levels within cardiovascular cells are the sum of fluxes through both the systemic and the local hepcidin/FPN axes [26,40]. A clearer understanding of the precise nature of the interplay between local and hepatic hepcidins in control of cardiovascular iron homeostasis is necessary to enable the development of strategies that can safely correct cardiovascular iron levels without impinging on systemic iron control. Such strategies, e.g., increasing cardiac hepcidin or decreasing cardiac FPN, could present a novel approach for the correction of myocardial iron deficiency. A number of human hepcidin agonists and FPN inhibitors are currently in clinical trials for the treatment of iron overload in hemochromatosis and thalassemia. It would be important to explore whether such compounds can correct myocardial iron deficiency in CHF patients. However, given the systemic role of hepcidin in the control of iron absorption and recycling, it may be more advantageous to target druggable differences between hepatic and cardiac hepcidins. Strategies for the treatment of iron deficiency in cardiovascular disease are outlined in Figure 1.

Iron deficiency affects cardiovascular function through anaemia-dependent and anaemia-independent pathways. Anaemia-independent pathways include direct effects of cardiac iron deficiency on metabolism, oxygen sensing, and calcium handling in the heart, and of vascular iron deficiency on oxygen sensing and vasoconstriction in the pulmonary arteries. Iron therapies for chronic heart failure and pulmonary arterial hypertension could be devised to directly correct local iron deficiency, enhance iron uptake (by enhancing local IRPs) and retention (by enhancing local HAMP, e.g., using bone morphogenetic protein receptor 2 (BMPR2) ligands for PAH), or to correct downstream changes in metabolism, oxygen sensing (e.g., HIF inhibitors), calcium handling, or vasoconstriction (e.g., endothelin-1 (ET-1) receptor antagonists).

## 2. Iron and the Pulmonary Vasculature

### 2.1. Effects of Iron Deficiency on the Pulmonary Vasculature

Systemic iron deficiency impinges on the pulmonary vasculature both in the context of normal physiological responses and in the setting of disease, e.g., pulmonary arterial hypertension (PAH). For instance, both clinical iron deficiency and acute infusion with the iron chelator deferoxamine augment the pulmonary vasoconstrictive response to hypoxia [42,43,44,45]. Iron deficiency is also prevalent in patients with PAH, in whom it is associated with reduced performance in the six-minute walk test (6MWT), increased disease severity, and poor clinical outcome [46,47,48]. It is also associated with higher pulmonary arterial pressure in other disease settings [49,50]. Rats made iron-deficient (and anaemic) through dietary iron restriction also develop PAH [51]. The mechanistic links between iron deficiency and PAH have remained unclear, due to the limitation posed by the existing experimental models, which do not discern the systemic effects of anaemia on tissue oxygenation and exercise capacity, from the local effects of iron deficiency directly acting in the pulmonary vasculature. However, a direct cause/effect relationship between vascular iron deficiency and PAH has recently been demonstrated in mice [52]. It was shown that mice with a smooth muscle-specific knock-in of FPN C326Y develop PAH as a consequence of intracellular iron deficiency in pulmonary arterial smooth muscle cells (PASMCs). Iron deficiency directly exerted its effects on PASMCs by increasing the expression of the endogenous vasoconstrictor endothelin-1 (ET-1) [52]. The roles of ET-1 in normal vascular responses to hypoxia and in the aetiology of PAH have long been recognised. ET-1 is elevated in the lung and in the circulation of PAH patients [53,54], and a number of trials have demonstrated the benefits of ET receptor antagonists in this disease [55]. The *edn1* gene is also a known HIF-regulated gene. The regulation of *edn1* by iron represents another example of the interplay between oxygen and iron homeostasis, and suggests that iron levels in the vascular bed may alter the magnitude of HIF-driven responses to hypoxia.

### 2.2. Targeting Iron Deficiency in Pulmonary Arterial Hypertension

Intravenous iron infusion has been shown to decrease the magnitude of the normal acute hypoxic response in healthy individuals, attenuate the exaggerated hypoxic response in iron-deficient individuals, and improve performance in the six-minute walk test (6MWT) in PAH patients [42,43,44,45,46,56,57]. The mechanisms underlying these effects are not entirely clear, but as with heart function, they likely involve indirect effects via improved exercise capacity and tissue oxygenation, and direct effects on iron-dependent pathways in the pulmonary vascular bed. The latter mechanism is supported by the finding that intravenous iron replenishes intracellular iron levels in PASMCs in mice, decreasing ET-1 release, and preventing and partially reversing the development of PAH [52]. The effects of intracellular iron levels on ET-1 levels were further recapitulated in-vitro in human PASMCs, suggesting a direct effect of iron on the expression of *edn1*, possibly involving blunting of the HIF pathway [52]. This “direct” mechanism is further supported by the finding that intravenous iron, administered immediately before acute hypoxic exposure, blunts the magnitude of the rise in PAP, before any changes in haemoglobin levels [43]. This is associated with an inhibition of the rise in serum ET-1 that otherwise accompanies acute hypoxia exposure [52]. These data give rise to the notion that iron levels in the pulmonary vasculature should be considered as a new target in the treatment of PAH. Existing iron preparations in the clinic have been developed to correct anaemia, but it would be important to explore whether such compounds can safely and efficiently replenish iron in the pulmonary vasculature without causing iron toxicity in the long term.

An alternative approach to directly delivering iron to the vascular tissue would be to target the molecular machinery of local cellular iron homeostasis. As with cardiomyocytes, IRPs appear to be involved in intracellular iron homeostasis in PASMCs. Mice lacking IRP1 have been shown to develop PAH through a mechanism involving the translational derepression of HIF-2α and subsequent increase in its target gene *edn1* [58]. More recently, the cell-autonomous control of intracellular iron through the autocrine action of the hepcidin/FPN axis was also demonstrated in PASMCs. The loss of such regulation was sufficient to cause PAH [52]. This study also provided evidence that the deregulation of this cell-autonomous pathway may be an aetiological factor in familial PAH. Indeed, PASMCs from patients with mutations in bone morphogenetic protein receptor 2 (*bmpr2*), which cause heritable PAH, were shown to have decreased hepcidin expression, increased FPN levels, reduced intracellular iron levels, and increased levels of ET-1. All of these effects could be reversed in-vitro by treatment with iron or exogenous hepcidin peptide [52]. These findings present an entirely novel mechanism through which *bmpr2* mutations may cause PAH. Therefore, targeting the hepcidin/FPN axis in PASMCs may hold therapeutic potential in the treatment of PAH. This would require identification of druggable differences between hepcidin derived from PASMC and hepatic hepcidin. One possibility is the selective enhancement of BMPR2 signalling in PASMCs through the use of specific BMPR2 ligands, which stimulate hepcidin in PASMCs without affecting hepatic hepcidin. Further studies are warranted to identify the mechanisms of hepcidin regulation downstream of BMPR2 in PASMCs. Strategies for the treatment of iron deficiency in PAH are outlined in Figure 1.

## 3. Conclusions

Iron deficiency is a recognised co-morbidity in several cardiovascular diseases. In chronic heart failure and pulmonary arterial hypertension, direct effects of iron deficiency within the cardiovascular tissue have been demonstrated, highlighting local iron deficiency as a new therapeutic target in these diseases. Some clinically used iron preparations appear to exert their benefits, in part, by the direct replenishment of intracellular iron levels in the cardiovascular tissue. Recent insights into the molecular machinery of cellular iron homeostasis in the heart and the pulmonary vasculature provide novel therapeutic targets. These targets hold the potential to correct local iron deficiency in the cardiovascular tissue without impinging on systemic iron control.

## Figures and Tables

**Figure 1 pharmaceuticals-12-00125-f001:**
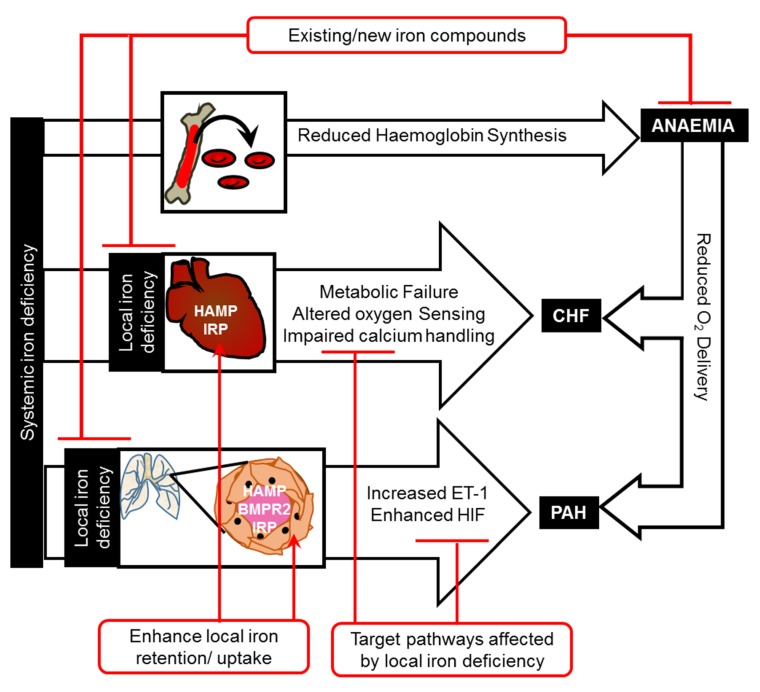
Mechanisms linking iron deficiency and cardiovascular disease.

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
