# Peer review of "Iron Deficiency as a Therapeutic Target in Cardiovascular Disease"

_pharmaceuticals, 2019, doi:10.3390/ph12030125_

Round 1

Reviewer 1 Report

The manuscript reviews a very interesting and important topics. It summarizes available data on the association of chronic heart failure with iron deficiency, and proposes theoretical strategies how to replete iron stores in heart failure patients.

The main concern is that the manuscript does not provide all necessary information to the reader; particularly it lacks information on the effect of the opposite situation, i.e. iron overload on cardiac physiology. In addition, the cited references are not in all cases relevant to the text.

Major point:

The manuscript strongly supports the general idea that iron supplementation is beneficial to patients with chronic heart failure. To present the reader with a whole picture, it should definitively also mention that, on the other hand, transfusional iron overload invariably leads to fatal heart failure, as documented in thalassemia major patients. Whereas iron supplementation might indeed be beneficial in heart failure patients, iron overload definitely IS causing heart failure. Actually, the author herself has shown that FPN deletion in cardiomyocytes leads to premature death of experimental animals, thus demonstrating the detrimental effect of iron overload on the heart. Iron is traditionally regarded as a doble-edged sword, and this must be mentioned in the manuscript.

The manuscript advocates the targeting of the cardiac hepcidin/ferroportin axis. To understand this strategy, the author should probably speculate on the putative mechanism by which cardiac ferroportin escapes the regulation by systemic hepcidin. In this context, it should be noted that hepatic hepcidin mRNA content is approximately by two orders of magnitude higher than cardiac hepcidin mRNA content. In her previous review published in 2019, the author states that it should be established how the cardiac-specific and systemic hepcidin/ferroportin axes interact; the present review gives the impression that it is primarily the cardiac-specific axis which controls cardiac iron metabolism. The interplay between systemic and cardiac-specific hepcidin synthesis should probably receive more attention in the manuscript.     

Minor points:

Abstract, first sentence: The statement that iron deficiency is particularly prevalent amongst patients with cardiovascular disease is probably exaggerated - iron deficiency is particularly prevalent among patients with chronic bleeding or nutritional deficiency etc. Please substitute "particularly".

Line 78-79: Sentence is (probably) incomplete

Line 82: Reactive oxygen species

Line 137-138: Sentence is incomplete

Line 150: Please use correct gene nomenclature (EDN1

Line 195: It is not clear why the text in this figure should mention cardiac iron deficiency

Reference 8: Does not mention the participation of  labile free iron (as suggested in the manuscript text)

Reference 11: Does not mention cardiomyocytes (as suggested in the text)

Reference 12: Does not mention cardiomyocytes (as suggested in the text) 

The References section should be carefully re-checked for typing errors and omissions (reference 24, reference 26...) 

Reviewer 2 Report

Iron deficiency as a therapeutic target in cardiovascular disease.

Lakhal-Littleton S.

 This review reports the possible role of iron in two cardiac pathologies: cardiac failure and pulmonary arterial hypertension.

1.    One figure with the potential role of iron, hepicin and ferroportin on systemic effects but specific effects on cardio vascular will probably make easier the explanations for the reader on the roles of iron. The two current figures could be summarized in one

2.    Therapies evoked by playing on the hepcidin-ferroportin axis are identical for both pathologies. It would be easier for the reader to make a single paragraph on potential treatments after explanations of the potential role of iron in the two pathologies.

3.    Page 2, line 31: “iron deficiency…..cause of anaemia”. A reference is needed.

4.    Page 2: iron deficiency. Please separate a real deficiency of iron from a functional deficit on an inflammation.

5.    Page 3 line 111. Define FPN

6.    Page 4: define HAMP

7.    Page 7: references are not well written (references 16, 24)

8.    Had perhaps reference from the NEJM to illustrate the role of HIF inhibitor (Chen N et al. NEJM 2019; July 24)

Reviewer 3 Report

General comments:

Well-written review about an interesting subject with potential to change our therapeutic approach in the future.

Specific comments:

Line 137-138: A study using dietary iron restriction in rats also found that such rats developed PAH associated (40).

I don't understand the word associated here: what does it mean? It is a strange sentence.

Round 2

Reviewer 1 Report

The author has adequately addressed all the points, with the sole exception of one remaining problem with the references cited:

Lines 39 to 41 state that labile free iron is required for the correct function of dioxygenases and ROS-producing enzymes. The wording of the sentence suggests that dioxygenases and ROS-producing enzymes can not function without some amount of labile iron present in the cell. Since labile free iron is traditionally regarded as detrimental, the statement represents an interesting viewpoint which should be supported by appropriate references. However, neither of the two references cited (17 and 18) mentions the labile iron pool and therefore does not support the statement. Please use other references (preferably directly addressing the role of labile iron in cell metabolism).      

Author Response

We thank the reviewer for raising this important point. The statement that the labile iron pool is required for the generation of ROS is based on the finding that ROS production is reduced by iron chelators. The widely accepted definition of the labile iron pool is that it is  a chelator-accessible source of reactive oxygen species.  In addition, the activity of HIF prolyl hydroxylases (from the family of oxygen and iron-dependent dioxygenases) is also inhibited by iron chelators, which accounts for the well-documented induction of HIF by iron chelators in vitro and in-vivo. This supports the idea that the labile iron pool is required for the activity of these enzymes. We now include additional references (18 and 20 in the revised manuscript) to support this statement.

Reviewer 2 Report

The author  responded to all my remarks. The manuscript is improved

Author Response

Thank you.